# Fiber-optic seismic sensing of vadose zone soil moisture dynamics

Zhichao Shen [1,2,5] ✉, Yan Yang [1,5], Xiaojing Fu[3], Kyra H. Adams [4], Ettore Biondi [1] & Zhongwen Zhan[1]

Vadose zone soil moisture is often considered a pivotal intermediary water reservoir between surface and groundwater in semi-arid regions. Understanding its dynamics in response to changes in meteorologic forcing patterns is essential to enhance the climate resiliency of our ecological and agricultural system. However, the inability to observe high-resolution vadose zone soil moisture dynamics over large spatiotemporal scales hinders quantitative characterization. Here, utilizing pre-existing fiber-optic cables as seismic sensors, we demonstrate a fiber-optic seismic sensing principle to robustly capture vadose zone soil moisture dynamics. Our observations in Ridgecrest, California reveal sub-seasonal precipitation replenishments and a prolonged drought in the vadose zone, consistent with a zero-dimensional hydrological model. Our results suggest a significant water loss of 0.25 m/year through evapotranspiration at our field side, validated by nearby eddy-covariance based measurements. Yet, detailed discrepancies between our observations and modeling highlight the necessity for complementary in-situ validations. Given the escalated regional drought risk under climate change, our findings underscore the promise of fiber-optic seismic sensing to facilitate water resource management in semi-arid regions.

The vadose zone plays a vital role in sustaining natural ecosystems[1–4], altering water and nutrient cycles[5–7], and informing agricultural water resource management[8,9]. Acting as an intermediary between surface soil moisture and deep groundwater, it is thought to function as a backup reservoir of water in semiarid regions, which is crucial for strengthening the resilience of our ecological and agricultural systems[1,10,11]. To the first order, water stored in the vadose zone can be readily replenished by precipitation and lost to the atmosphere via evapotranspiration or deep drainage to recharge groundwater (Fig. 1a). While this conceptual model has long been indoctrinated in hydrology, it is challenging to quantitatively monitor the long-term, large-scale vadose zone soil moisture dynamics at depth. Modern microwave remote sensing missions, such as the SMAP[12] (Soil Moisture Active Passive) and SMOS[13] (Soil Moisture and Ocean Salinity), and

GNSS (Global Navigation Satellite System) based techniques[14,15] provide good estimates of global soil moisture only down to ~5 centimeters at 10–40 km spatial resolution every few days and can further extend to the 1-m root zone using data assimilation[16]. Satellite-based gravimetric measurements are sensitive to greater depths but rely on aforementioned microwave missions to disaggregate between surface soil moisture and deep groundwater[17]. In situ, instruments such as lysimeters allow for direct measurements of the water content change in unsaturated soil, which are commonly used as a tool to calibrate satellite-derived soil moisture[18]. Yet, these point-wise measurements do not provide a large-scale view of vadose zone dynamics. Other hydro-geophysical means, including time-domain reflectometer, ground penetrating radar, and electromagnetic system, can characterize detailed vadose zone soil structures[19–22] but the operational

[1]Seismological Laboratory, California Institute of Technology, Pasadena, CA, USA. [2]Department of Geology and Geophysics, Woods Hole Oceanographic Institution, Woods Hole, MA, USA. [3]Department of Mechanical and Civil Engineering, California Institute of Technology, Pasadena, CA, USA. [4]Jet Propulsion Laboratory, California Institute of Technology, Pasadena, CA, USA. [5]These authors contributed equally: Zhichao Shen, Yan Yang. ✉e-mail: zhichao.shen@whoi.edu

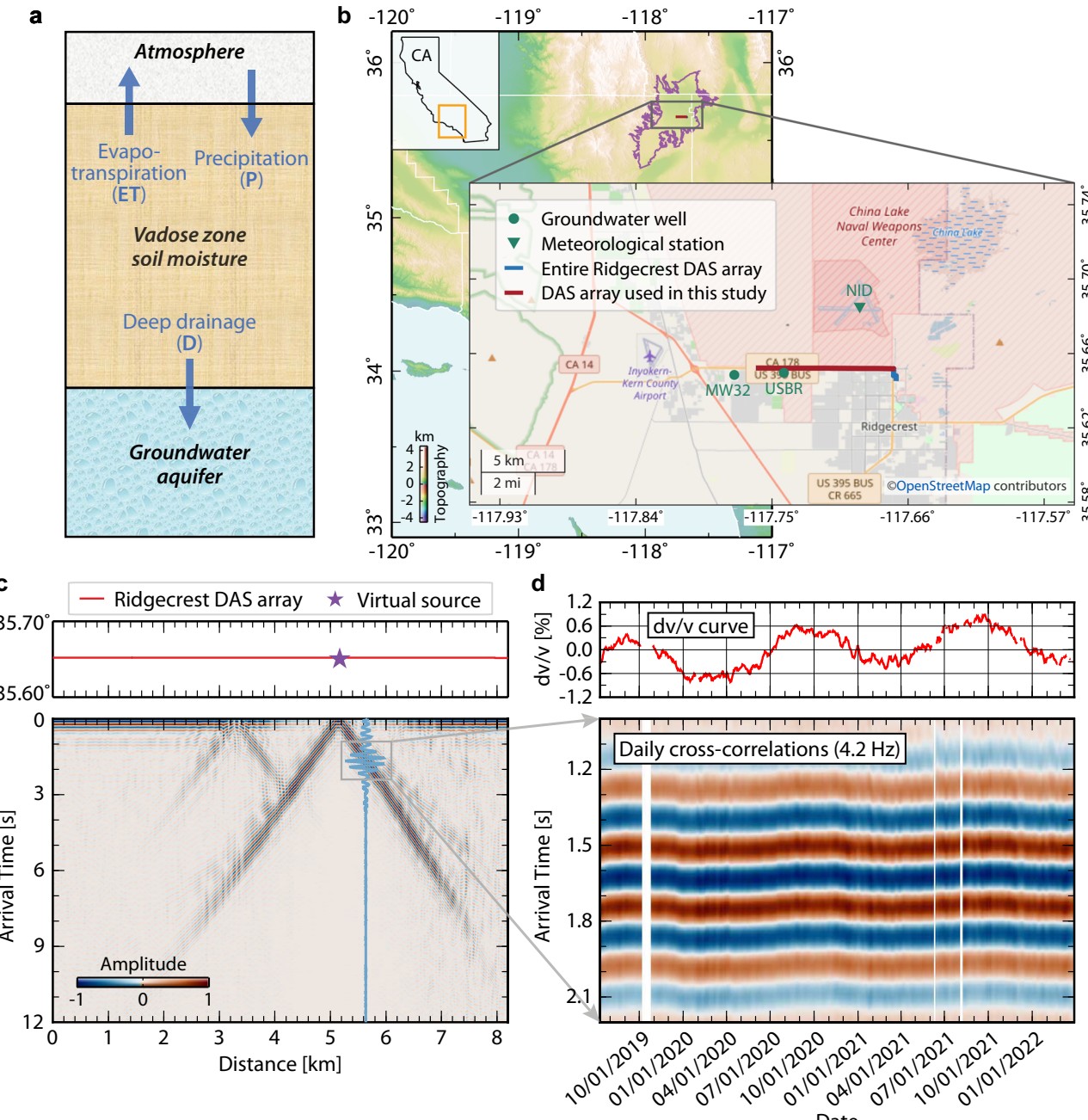

**Fig. 1 | Conceptual model for the vadose zone water dynamics and time-lapse seismology example on Ridgecrest DAS array. a** Vadose zone soil moisture can be readily replenished by precipitation (P) and lost due to evapotranspiration (ET) to the atmosphere or deep drainage (D) to recharge deeper aquifers. **b** Map view of Ridgecrest DAS array. The upper-left inset panel depicts the California (CA) map with Indian Wells Valley marked by a yellow box. The purple line represents the boundary of Indian Wells Valley. The lower-right inset panel shows a zoom-in view of the Ridgecrest DAS array, a nearby meteorological station (green triangle), and groundwater wells (green circles). The blue line and overlying red line represent the entire DAS cable and an 8-km DAS segment used in this study, respectively. The underlying street map is available from OpenStreetMap under the Open Database License (https://www.openstreetmap.org/copyright). **c** Record a section of weekly stacked seismic waveforms at 4.2 Hz from a virtual source (purple star) at a cable distance of 5.2 km. The blue waveform shows the seismogram received 60 channels away from the virtual source. **d** dv/v measurements (top panel) for the channel pair in (**c**). The bottom panel presents the temporal variation of direct surface waves. Vertical white strips are data gaps. Source data are provided as a Source Data file.

cost of long-term monitoring is often prohibitive for extensive deployment[16,23].

Recently, time-lapse seismology has shown great promise to characterize subsurface hydrological processes using the seismic velocity change (dv/v) as an indicator of water saturation[24–27]. Water content in the subsurface perturbs seismic velocity and thus can be inferred by repeatedly measuring the arrival time variation of seismic surface waves propagating between two seismic stations on a regular basis. In addition, the dispersive nature of surface waves with long periods sampling greater depths enables a glimpse into the depth-dependence of the subsurface water cycle. However, due to the sparsity of regional seismic networks in tens of kilometers spacing, time-lapse seismology typically has limited spatial resolution and uses long-period surface waves (i.e, 1–10 s) that are sensitive to groundwater rather than vadose zone soil moisture[25,26]. Therefore, resolving the water dynamics in the vadose zone is a task better suited for a denser continuous seismic network.

Distributed acoustic sensing (DAS) offers an affordable and scalable solution for long-term monitoring through ultra-dense seismic arrays[28]. By converting Rayleigh-backscattered laser signals due to intrinsic fiber impurities to longitudinal strain or strain rates, DAS repurposes pre-existing telecommunication fiber-optic cables into thousands of seismic sensors over tens of kilometers. With meter-scale channel spacing, DAS continuously records high-frequency wavefields and boosts the spatiotemporal resolution of time-lapse seismology at shallow depths[29]. Here, we demonstrate that the vadose zone soil moisture can be fiber-optically sensed using a DAS array in Ridgecrest, California.

Located to the north of the Mojave Desert, the city of Ridgecrest is nestled in the Indian Wells Valley basin (Fig. 1b), which is a typical semi-arid region receiving an annual precipitation of ~0.05 m (Supplementary Table 1). Due to the severe droughts in the past few years, groundwater in this region has been critically overdrawn to meet the agricultural and municipal demand. While this historic drought is reflected in the declining water levels in regional aquifers and the drying surface moisture[30], its impact on water stored in the shallow subsurface remains unclear. On July 10, 2019, a pre-existing telecommunication fiber optic cable along U.S. Route 395 Business, a major road in Ridgecrest, was transformed into a 10-km long DAS array with 1250 channels spacing 8 m (Fig. 1b). The Ridgecrest DAS array was initially deployed as a rapid response to the 2019 M7.1 Ridgecrest earthquake[31] and has been continuously running for two and a half years (Jul. 2019–Mar. 2022) with a few small data gaps.

In this study, we leverage time-lapse seismology and rock physics to develop a fiber-optic seismic sensing principle that passively and robustly captures the vadose zone soil moisture dynamics. Applying this technique to the Ridgecrest DAS array, we observe sub-seasonal precipitation replenishments and a prolonged drought in the vadose zone lasting over 2.5 years, consistent with a zero-dimensional hydrological model. Our findings reveal a significant water loss of 0.25 m/year through evapotranspiration in Ridgecrest, validated by eddy-covariance based measurements from a nearby region. Yet, regarding detailed vadose zone water dynamics, there are discrepancies between our observations and hydrological modeling, highlighting the need for complementary in situ validations. Nonetheless, our results demonstrate the promise of fiber-optic seismic sensing as a large-scale and long-term observational tool to facilitate water resource management in the face of escalating regional droughts.

## Results
### Observation of vadose zone soil moisture dynamics
The fine channel spacing of DAS enables the use of direct high-frequency surface waves for vadose zone monitoring. As an example, Fig. 1c clearly shows 4.2 Hz Rayleigh waves propagating along the array for ~2 km, derived with one week of ambient noise data (see "Methods" for details). For a channel pair separated by 480 m, repeated retrieval of surface waves weekly presents distinct signals of high signal-to-noise ratios with temporal variations of direct arrivals (Fig. 1d). Taking the 2.5-year averaged waveform as a reference, we calculate the relative seismic velocity change (dv/v) with time as the opposite of the relative time shift of surface wave arrivals. The resulting dv/v curve shows a clear seasonality with lows in the winter and highs in the summer, along with an upward trend over the 2.5-year observational period (Fig. 1d). Specifically, the annual highs of dv/v elevated from 0.4% in 2019 to 0.6% in 2021 and further to 0.8% in 2022 (Fig. 1d).

Repeating and integrating the measurements for all channel pairs, the 8-km-long DAS array reveals the spatiotemporal evolution of subsurface dv/v in unprecedented detail (Fig. 2; see "Methods" for details). We observe an evident spatial variation in dv/v amplitude with ±1.5% at the east end gradually decaying westward to ±0.5% at 2–3 km, followed by a slight increase at 0–2 km (Fig. 2b). This lateral spatial

variation correlates well with the tapering of the shallow sediment thickness westward from 60 m to less than 20 m (Fig. 2a). A similar correlation between larger dv/v and thicker sediment has been reported previously[25,26] but is observed on a much finer scale here. Moreover, we observe temporal variations of dv/v across multiple time scales. First, rapid dv/v amplitude drops are observed throughout the DAS array in response to sub-seasonal meteorological forcing (e.g, rainfalls in April 2020; Fig. 2c), leaving distinct horizontal footprints on the dv/v map (Fig. 2b). Second, our 4.2 Hz dv/v measurements exhibit strong seasonality and a long-term increasing trend along the cable (i.e, lighter red and deeper blue in Fig. 2b). In fact, rather than just a single frequency band, the cable-wide-averaged dv/v curves present consistent multiscale temporal variations across a broad frequency band (2.5–7.1 Hz; Fig. 3b), corresponding to a sensitivity depth extending down to 150 m (Fig. 3a and Supplementary Fig. 1). The dv/v amplitude monotonically increases with frequency, indicating a primary dv/v contribution from the top 20 m (Fig. 3a).

Among the observed dv/v variations over multiple time scales, the seasonal variation is the predominant one. Previous studies attribute seasonality in long-period seismic signals to groundwater level changes, but their reported dv/v amplitudes are one order of magnitude smaller than this study[25,26]. Such amplitude discrepancy arises from the choice of frequency band and inter-station distance. In fact, previous studies using high-frequency seismic waves yield the same order of magnitude variation in dv/v amplitudes[32–34]. Nonetheless, the groundwater table at our field site is approximately at depths of 75–90 m (Supplementary Fig. 2), far below the observable depth (top 20 m) of our strongest signals, therefore ruling out groundwater fluctuation as the main cause of our observed dv/v seasonality. On the other hand, surface temperature follows a seasonal pattern (Fig. 3c), and the associated thermoelastic effect can extend considerably deep[35–37]. Indeed, the surface temperature variation has been shown to correlate with the seasonality of dv/v amplitude comparable to our observations[33,38], suggesting thermal fluctuation as the main cause for the observed seasonality here. We correct the thermoelastic component of dv/v by scaling and shifting the surface temperature curve to best fit seasonal dv/v variations (Fig. 3c, see "Methods" for details). The estimated phase lags between temperature and our dv/v observations range from 10 to 60 days with longer delay time at greater depths, yielding a reasonable estimate of soil thermal diffusivity[33,35] (see "Methods" for details and Supplementary Fig. 3).

With the thermoelastic component corrected, the remaining cable-wide-averaged dv/v curves preserve sub-seasonal variations aligned with meteorological forcing and a frequency-dependent long-term trend (Fig. 3d). Specifically, we observe that intense rainfall sequences in January and April 2020 have led to dramatic dv/v drops of up to 1% within days, followed by a gradual recovery to elevated levels months later (Fig. 3d). Qualitatively, this can be explained as a direct result of soil moisture changes in response to meteorological forcing. As rainfalls recharge the vadose zone, increases in water saturation in soil perturb the elastic properties of porous soil[39,40], and consequently reduce the seismic velocity[32,41,42]. In contrast, outside of the wet season, vadose zone dynamics are dominated by water loss through evapotranspiration, resulting in a reduction in soil moisture and a recovery of the seismic velocity[32]. The soil drying process is especially prominent during the historic drought beginning in the summer of 2020[43], where we observe a sharp climb of dv/v (Fig. 3d). Overall, the long-term dv/v increase is consistently observed across a broad frequency band with a more pronounced effect at higher frequencies, implying that shallow soil moisture dynamics is the dominant driver to the observed seismic signals (Fig. 3a).

### Quantification of vadose zone soil moisture dynamics
To further quantify the soil moisture dynamics (i.e., time series of soil moisture budget) using dv/v observations, we resort to an

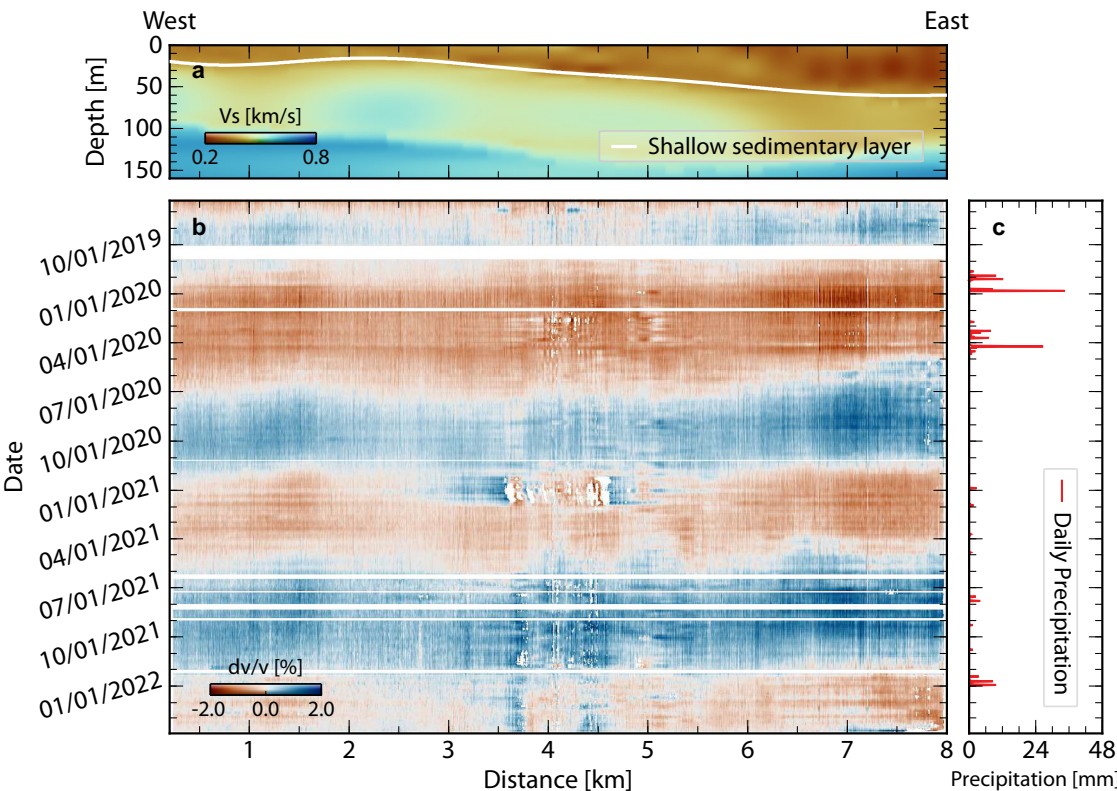

**Fig. 2 | Subsurface dv/v spatiotemporal evolution. a** Shear wave tomography beneath the Ridgecrest DAS array[52]. The white line delineates the shallow sedimentary layer. **b** 4.2 Hz dv/v results across the Ridgecrest DAS array. The white regions are either data gaps or low-quality dv/v measurements. Note the correlation between lateral variations of sediment thickness and dv/v measurements. **c** Precipitation data from meteorological station. Note the correlation between precipitation in (**c**) and horizontal dv/v anomalies in (**b**). Source data are provided as a Source Data file.

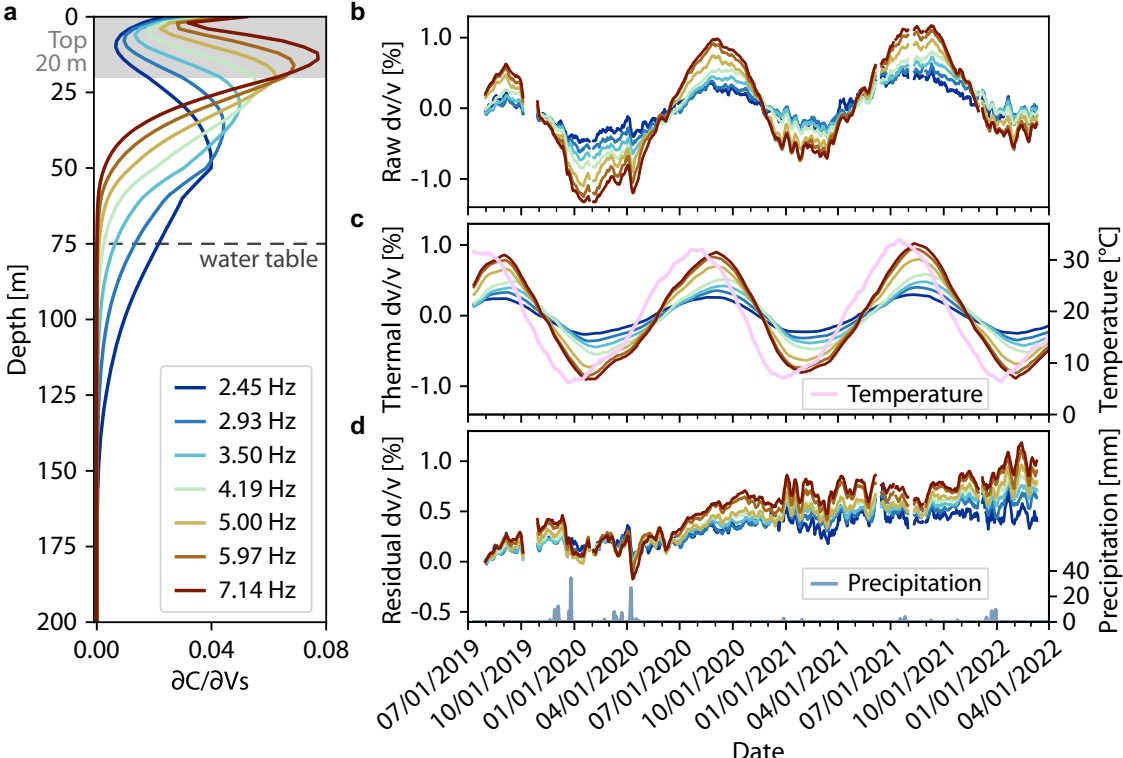

**Fig. 3 | Subsurface dv/v analysis at varying frequencies. a** Surface wave sensitivity kernels for different frequencies varying from 2.45 to 7.14 Hz. The black dashed line indicates the groundwater table, and the shaded area denotes the top 20 m. **b** Cable-wide-averaged dv/v curves for varying frequencies. **c** Thermal dv/v fitted by the surface temperature curve in pink. **d** Residual dv/v measurements (set to 0 on day 1 for visualization) after the thermoelastic correction. The blue line denotes precipitation. Source data are provided as a Source Data file.

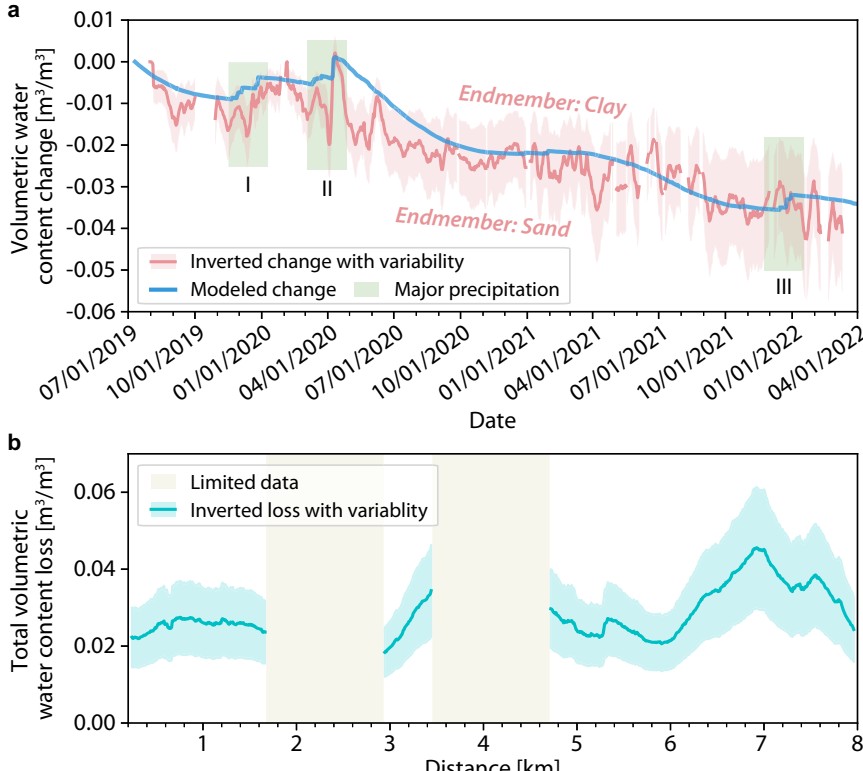

**Fig. 4 | Vadose zone soil moisture loss. a** Modeled (blue) and inverted (red) volumetric water content changes in the top 20 m at a cable distance of 7.2 km. The red-shaded area is bounded by inversion results derived from two end-member lithology models, and the red curve represents the mean variation. The three green-shaded areas highlight the major precipitation periods during our study period. **b** Lateral variation of total vadose zone soil moisture loss during the 2.5-year study period. The turquoise-color-shaded region indicates the variability in total soil moisture loss, bounded by the two end-member lithology models, with the solid line representing the mean variation. The light-gray shaded gaps indicate cable sections lacking sufficient high-quality data for robust estimates. Source data are provided as a Source Data file.

existing rock physics model[44–47] tailored for unconsolidated unsaturated porous soil medium (see "Methods" for details). The forward model calculates dv/v as a function of soil moisture profile and a given lithology model, which enables us to invert for water content changes in the top 20 m that can best fit dv/v measurements across all frequencies (see "Methods" for details and Supplementary Figs. 4a, 5). As the inversion requires an assumed lithology model, we examine two endmember lithology[44]—Esperance sand and Missouri clay—to establish upper and lower bounds for the inferred vadose zone water content changes. Despite amplitude differences, results from the two endmembers consistently mirror our observed dv/v patterns, capturing episodic replenishments from precipitation and a long-term trend of water loss from 2019 to 2022 (Fig. 4a).

For independent validation, we further use a zero-dimensional hydrological model forced with meteorological and SMAP data products to simulate subsurface volumetric water content changes at our field site[48,49] (Fig. 1a; see "Methods" for details and Supplementary Fig. 4b). Taking the eastern side of the DAS cable as an example, our modeled soil moisture changes closely track the dv/v inverted trend within the bounds of the two endmember cases and exhibit remarkable consistency with the mean curve (Fig. 4a). This robust quantitative consistency between our observation and modeling demonstrates the efficacy of fiber-optic seismic sensing in capturing vadose zone soil moisture dynamics. Our result unveils a significant water loss rate of 0.25 m/yr through evapotranspiration, surpassing the annual precipitation in Ridgecrest (Supplementary Table 1). In fact, the inferred value is consistent with the eddy-covariance based measurement of evapotranspiration of 0.2 m/yr in the nearby southwestern Mojave

Desert covered by similar vegetations[50] (see "Methods" for details). Furthermore, leveraging our dv/v data across the entire cable, we invert the lateral variations in total soil moisture loss over the 2.5-year observational period (see "Methods" for details; Fig. 4b). Despite variations in the assumed lithology models, the eastern end (i.e., 6-8 km) appears to experience a greater loss in vadose zone soil moisture compared to the western end (Fig. 4b), suggesting a high sensitivity of evapotranspiration rate to localized environmental forcing. Achieving such high-resolution spatial variations is challenging with satellite-based remote sensing techniques, but feasible with our fiber-optic seismic sensing. Extrapolating the spatially averaged water content change in our field site to the entire Mojave Desert region, a first-order estimate yields an annual water loss of ~30 km³, comparable to the total capacity of Hoover Dam.

## Discussion

Despite the overall agreement between our modeled and inverted water content changes, quantitative discrepancies still exist in finer details. Unlike the rapid replenishment from the two rainfall sequences in 2020, our inverted declining trend of soil moisture is not perturbed by the precipitation sequence in December 2021, differing from the hydrological model (Fig. 4a). This discrepancy may arise from oversimplification of the hydrological model, which does not consider surface runoff. Moreover, the modeled soil moisture loss rate between April and October 2020 is slower than our observed rate (Fig. 4a), likely due to an underestimation of evapotranspiration in our hydrological model. To this extent, our dv/v observations could impose tight constraints on parameterized hydrological models at the regional scale.

With the Ridgecrest DAS array, we demonstrate the viability of fiber-optic seismic sensing for high-resolution vadose zone soil moisture dynamics. For practical applications of fiber-optic seismic sensing in various environments, the noise source distribution, poor cable coupling, and unclear subsurface lithology would lead to uncertainties in the fiber-optically estimated vadose zone water content changes. Combined with continued seismic sensing, future work involving direct in situ measurements for different soil types and regional scale modeling will increase the robustness of the inverted vadose zone soil moisture dynamics. Given the escalated regional drought risk under climate change, our findings highlight the promise of fiber-optic seismic sensing as a large-scale, long-term, and cost-effective observational tool to enhance our climate resilience in semi-arid regions.

## Methods

### Time-lapse seismology

To compute ambient noise cross-correlations, we preprocess daily seismograms by removing mean and linear trend, bandpass filtering between 1–10 Hz, downsampling to 50 Hz, time domain normalization, and spectral whitening[51]. The 24-h seismic data are then cut into 40-s segments for each channel. For any given channel pair, all the segments are cross-correlated, normalized, and stacked to present the daily cross-correlation. The 2.5-year continuous raw data recorded by our Ridgecrest DAS array yields a total volume of 80 TB for ambient noise cross-correlations. We implement GPU-based parallel processing to effectively accelerate the massive computation. Once the daily cross-correlations are computed, we smooth them over a seven-day moving window to reduce periodic traffic source effect[52]. The causal and acausal branches of cross-correlations show great symmetry (Supplementary Fig. 6), suggesting the distribution of noise sources is overall even. We also observe clear scattered surface waves at ~4 km, which have been suggested as the seismic signatures of a geologically inferred fault[53]. To further enhance the signal-to-noise ratio (SNR), we average the causal and acausal branches for all the cross-correlations.

We use the direct surface wave to assess seismic velocity changes (dv/v) rather than coda waves[54–56] because its sensitivity kernel is more deterministic. To quantify the seismic velocity change (dv/v), we apply a cross-spectrum method to compute the time shifts of 1-s surface wave windows between daily and reference cross-correlations[57]. Here, we use the cross-spectrum method for its extensive use and effective uncertainty quantification[25,57]. Benefitting from our high-quality cross-correlations, the resulting dv/v uncertainties are generally small, with the majority falling below 0.04% (Supplementary Fig. 7). Nonetheless, notable large dv/v uncertainties are observed near the 4-km mark along the cable (vertical stripes in Supplementary Fig. 7), which is attributed to scattered surface waves affecting the signal quality of retrieved direct surface waves (Supplementary Fig. 6). Other methods, such as a straightforward time-domain cross-correlation and wavelet-based approaches, have also demonstrated great performance in previous dv/v studies[58,59]. The use of these methods for DAS data under various signal-to-noise levels remains further investigation to reduce the uncertainty. The reference waveform is obtained by averaging over all the daily cross-correlations and the 1-s window is chosen as 0.5 s before and after the peak of the reference surface wave. Our dv/v results are opposite to the time shift measurements and are only accepted for waveform coherency larger than 0.5 and dv/v uncertainty smaller than 0.1%.

Rolling along the 8-km segment, we repeat the above procedures for all source-receiver pairs that are 480 m (60 channels) apart and pinpoint the dv/v results to their center locations, resulting in a time-lapse dv/v map of the Ridgecrest DAS array. An inter-channel distance of 480 m satisfies the three-wavelength criterion for high-frequency (> 2 Hz) Rayleigh waves, thus sufficient to develop robust waveforms for direct arrivals[51].

We use Rayleigh waves at a broad frequency band of 1–10 Hz and 7 sub-bands centered at 2.45, 2.93, 3.50, 4.19, 5.00, 5.97, and 7.14 Hz for dv/v, which allows us to constrain temporal changes at depths. Although our shear wave velocity model achieves high resolution in the top 150 m, the velocity uncertainty at very shallow depths remains high[52]. Future attempts incorporating high-quality higher-frequency ballistic waves would enhance the depth resolution of our tomography model. Thus, we calculate Rayleigh wave sensitivity kernels based on the 1D shear velocity model averaged along the cable[52]. The observed dv/v increases with frequency, reflecting an exponentially decaying sensitivity of surface waves with depth[55,56,58,59] and suggesting that the major contribution comes from the top 20 m (Supplementary Fig. 1).

### Correction for thermoelastic effects

We follow the framework proposed by Richter et al.[38], which has a straightforward functional form:

$$\left(\frac{\mathrm{d}v}{v}\right)_{\mathrm{thermo}} = a\delta T(t - \tau) \tag{1}$$

where $\delta T(t)$ is the demeaned daily surface air temperature at time $t$. Our objective is to determine the amplitude $a$ and phase shift $\tau$ by optimization techniques. To focus on the annual pattern related to temperature, we use a sliding window average method to remove the transient signals. For each frequency, we use grid search to find the best length of the sliding window between 0 and 90 days, which can maximize the correlation coefficient between the smoothed dv/v and surface temperature. We then use grid search to find the best-fitted parameters $(a, \tau)$ that minimize the difference between the observed dv/v and the functional form $a\delta T(t - \tau)$. By removing thermoelastic effects from the dv/v, we tease out a more accurate representation of the subsurface velocity changes that are related to soil moisture fluctuations.

The thermoelastic induced dv/v changes also show depth-dependent patterns. With increasing frequency, the maximum correlation coefficient is larger, and the fitted $\tau$ is smaller (Supplementary Fig. 3), implying greater thermoelastic effects and more synchronization with temperature fluctuations at shallower depths. The observed amplitude of thermoelastic dv/v has a comparable amplitude with previous studies[33,38,60]. We can further use the time delay between dv/v and temperature to calculate the thermal diffusivity following the thermoelastic modeling[61] with a thermal thickness of 2 m. Consequently, this yields an average value of $1.49 \times 10^{-6}\,\mathrm{m^2/s}$. consistent with previously inferred values[33,35,61].

### Estimating vadose zone water loss from dv/v observations

To establish the quantitative relations between vadose zone soil moisture and dv/v, we built on and modified the theoretical framework proposed by Solazzi et al.[44]. Here, we briefly summarize the Solazzi model that enables a direct prediction of dv/v from a given vadose zone soil moisture change, which follows dv/v(t) = f(vadose zone soil moisture change, lithology model, t) (Supplementary Fig. 4a). Assuming that the vadose zone consists of $n$ unconsolidated soil layers of isotropic soil property with a water saturation $s_j$ and a thickness of $h_j$ where $j$ denotes the $j$th layer, the P and S wave velocity of the $j$th layer are given as[62]

$$\mathbf{V}_{P,j} = \sqrt{\frac{\mathbf{K}_j + \frac{4}{3}\boldsymbol{\mu}_j}{\boldsymbol{\rho}_j}}; \mathbf{V}_{S,j} = \sqrt{\frac{\boldsymbol{\mu}_j}{\boldsymbol{\rho}_j}} \tag{2}$$

where $\mathbf{K}_j$, $\boldsymbol{\mu}_j$ and $\boldsymbol{\rho}_j$ represent the effective bulk modulus, effective shear modulus, and effective bulk density in the $j$th layer, respectively. To account for the saturating water effect, we adopt the classic Biot-

Gassmann equations[46] to compute the effective elastic moduli and bulk density.

$$\mathbf{K}_j = \mathbf{K}_{d,j} + \frac{\left(1 - \frac{\mathbf{K}_{d,j}}{\mathbf{K}_{Se,j}}\right)^2}{\frac{\boldsymbol{\phi}_j}{\mathbf{K}_{f,j}} + \frac{1 - \boldsymbol{\phi}_j}{\mathbf{K}_{Se,j}} - \frac{\mathbf{K}_{d,j}}{\mathbf{K}_{Se,j}^2}} \tag{3}$$

$$\boldsymbol{\mu}_j = \boldsymbol{\mu}_{d,j} \tag{4}$$

$$\boldsymbol{\rho}_j = \left(1 - \boldsymbol{\phi}_j\right)\boldsymbol{\rho}_{Se,j} + \boldsymbol{\phi}_j\left[\mathbf{s}_j\boldsymbol{\rho}_w + \left(1 - \mathbf{s}_j\right)\boldsymbol{\rho}_a\right] \tag{5}$$

where $\mathbf{K}_{f,j}$, $\mathbf{K}_{Se,j}$, and $\mathbf{K}_{d,j}$ denote the fluid bulk modulus, effective bulk modulus of solid grains, and drained bulk moduli of the porous medium in the $j$th layer, respectively. $\boldsymbol{\phi}_j$ and $\boldsymbol{\mu}_{d,j}$ are the porosity and drained shear moduli of the $j$th layer, respectively. $\boldsymbol{\rho}_{Se,j}$, $\boldsymbol{\rho}_w$ and $\boldsymbol{\rho}_a$ represent the density of solid grains, water, and air, respectively. For low-frequency seismic waves, the fluid bulk modulus $\mathbf{K}_{f,j}$ can be approximated as:

$$\mathbf{K}_{f,j} = \left(\frac{\mathbf{s}_j}{\mathbf{K}_w} + \frac{1 - \mathbf{s}_j}{\mathbf{K}_a}\right)^{-1} \tag{6}$$

where $\mathbf{K}_w$ and $\mathbf{K}_a$ are the bulk moduli of water and air, respectively. Based on the classic Hertz-Mindlin theory[45], we compute the drained elastic moduli $\mathbf{K}_{d,j}$ and $\boldsymbol{\mu}_{d,j}$ given as:

$$\mathbf{K}_{d,j} = \left[\frac{N^2\left(1 - \boldsymbol{\phi}_j\right)^2\boldsymbol{\mu}_{Se,j}^2}{18\pi^2\left(1 - \boldsymbol{\nu}_{Se,j}\right)^2}\mathbf{P}_{e,j}\right]^{\frac{1}{3}} \tag{7}$$

$$\boldsymbol{\mu}_{d,j} = \frac{2 + 3f - (1 + 3f)\boldsymbol{\nu}_{Se,j}}{5(2 - \boldsymbol{\nu}_{Se,j})}\left[\frac{3N^2(1 - \boldsymbol{\phi}_j)^2\boldsymbol{\mu}_{Se,j}^2}{2\pi^2(1 - \boldsymbol{\nu}_{Se,j})^2}\mathbf{P}_{e,j}\right]^{\frac{1}{3}} \tag{8}$$

where $N$ and $f$ are the average number of contacts per particle and the fraction of non-slipping particles. $\mathbf{P}_{e,j}$ denotes the effective pressure. Solazzi et al.[44]. implemented a water saturation- and depth-dependent $\mathbf{P}_{e,j}$ to investigate the capillary suction effect on the seismic velocity and suggested that the surface-wave dispersion in coarse-grained soil textures is not sensitive to capillary effects, which are consistent with both laboratory[63,64] and field observations[65]. In particular, during the pendular stage, the capillary force can perturb small strain stiffness but may not significantly affect large strain stiffness[64]. Since our study region predominantly consists of coarse-grained sand soil, we opt out of including the capillary effect in the calculation of $\mathbf{P}_{e,j}$. $\boldsymbol{\nu}_{Se,j}$ and $\boldsymbol{\mu}_{Se,j}$ are Poisson's ratio and the effective shear modulus of the solid grains in the $j$th layer, respectively. Assuming the soil at the $j$th layer consists of $m$ types of constituents, we employ Hill's equation[47] to compute $\boldsymbol{\nu}_{Se,j}$, $\boldsymbol{\mu}_{Se,j}$ and $\boldsymbol{\rho}_{Se,j}$.

$$\boldsymbol{\nu}_{Se,j} = \frac{3\mathbf{K}_{Se,j} - 2\boldsymbol{\mu}_{Se,j}}{2(3\mathbf{K}_{Se,j} + \boldsymbol{\mu}_{Se,j})} \tag{9}$$

$$\boldsymbol{\mu}_{Se,j} = \frac{1}{2}\left[\sum_{i=1}^{m}\boldsymbol{\gamma}_{i,j}\boldsymbol{\mu}_{i,j} + \frac{1}{\sum_{i=1}^{m}\frac{\boldsymbol{\gamma}_{i,j}}{\boldsymbol{\mu}_{i,j}}}\right] \tag{10}$$

$$\boldsymbol{\rho}_{Se,j} = \sum_{i=1}^{m}\boldsymbol{\gamma}_{i,j}\boldsymbol{\rho}_{i,j} \tag{11}$$

where $\boldsymbol{\gamma}_{i,j}$ and $\boldsymbol{\rho}_{i,j}$ are the volumetric fraction and density of the $i$th constituent in the $j$th layer. $\mathbf{K}_{Se,j}$ denotes the effective bulk modulus of

the corresponding grains in the $j$th layer, which is given as

$$\mathbf{K}_{Se,j} = \frac{1}{2}\left[\sum_{i=1}^{m}\boldsymbol{\gamma}_{i,j}\mathbf{K}_{i,j} + \frac{1}{\sum_{i=1}^{m}\frac{\boldsymbol{\gamma}_{i,j}}{\mathbf{K}_{i,j}}}\right] \tag{12}$$

Following Solazzi et al.[44]. we selected parameters for Esperance sand and Missouri clay to represent two end-member lithology models (Supplementary Table 2). These models serve as the upper and lower bounds for our estimated volumetric water content. We also calculated the average of these two models, which we compared with the outputs from the hydrological model.

Based on the forward modeling from water saturation to dv/v, we conduct a grid search approach for the daily vadose zone soil moisture change (**s**) to best fit our daily frequency-dependent dv/v observations (Supplementary Fig. 4a). Typically, the number of vadose zone soil layers $n$ is greater than our dv/v observations (i.e, dv/v data at 7 center frequencies), making the inverse problem underdetermined. To simplify the problem, we assume that the daily vadose zone water content change primarily comes from the surface and follows an exponential decay with depth.

$$\Delta\mathbf{s}(z) = \Delta s_0 e^{-\frac{z}{\lambda}}, \tag{13}$$

where $\Delta s_0$ is the magnitude and $\lambda$ is an exponential decay parameter. To justify, our chosen mathematical form aligns with theoretical principles, as it mirrors the exponential profile typically observed in grass roots[66,67], which strongly governs the exponential decay in evapotranspiration with increasing depth. In this manner, the simplified vadose zone water content profile significantly reduces the model space to be searched and enables a straightforward comparison with our hydrological modeling. Given a specific lithology model, we calculate the elastic moduli before and after changes in water content. This model allows us to determine the depth-dependent seismic velocity changes before and after these water content alterations and, consequently, to compute the frequency-dependent dv/v. We fix $\lambda = 1\,\mathrm{m}$ based on the average root depth in this region and perform a grid search for $\Delta s_0$ ranging from $-0.1\,\mathrm{m^3/m^3}$ to $0.1\,\mathrm{m^3/m^3}$ with an interval of $0.001\,\mathrm{m^3/m^3}$ to minimize the mean-square misfit between predicted and observed dv/v across different frequencies for a 400-m cable segment at the eastern end (i.e., 7.2–7.6 km). For instance, given a lithology model averaged from the two endmembers (red line in Fig. 4 and Supplementary Table 2), a soil moisture increase of $0.0275\,\mathrm{m^3/m^3}$ in the top 20-m vadose zone effectively explains the observed trend in dv/v, which monotonically varies from $-0.23\%$ at 2.45 Hz to 1.24% at 7.14 Hz following the April 2020 precipitation event (Supplementary Fig. 5).

Repeating the inversion of daily volumetric water content changes for all DAS channels, we can estimate the total vadose zone soil moisture loss across the 8-km cable over the 2.5 years of our observational period. To ensure robust estimates, we apply the following two criteria for data quality control: (a). For a given channel and a specific date, the square root of the resolved mean square misfit must be less than 0.1%; (b). For a given channel, the number of inverted daily soil moisture changes must exceed 70% of the total days of our observation period to robustly determine total soil moisture loss. To account for lithology model uncertainties, we use the two end-member models as estimate bounds to represent the variability of the total soil moisture loss. The resulting lateral variations of total vadose zone soil moisture loss are shown in Fig. 4b, with two gaps at the 2–3 km and around 4-km mark. The gaps are due to insufficient high-quality data that can meet our criteria. For example, the dv/v uncertainty is generally large around the 4-km mark (Supplementary Fig. 7).

## Hydrological modeling

We use a physics-based 0-D bucket model to describe the water mass balance in the top 20 m subsurface (Supplementary Fig. 4b). Specifically, the inputs include precipitation and temperature data from the local meteorological station NID (Fig. 1), and surface soil moisture from SMAP[49]. Following Stahl and McColl[48], the water balance of a vertically averaged, horizontally homogeneous control volume of soil extending from the land surface down to a depth $\Delta z$ [m] (Fig. 1a) can be modeled as:

$$\Delta z \left( \theta_{fc} - \theta_w \right) \frac{d\mathbf{s}}{dt} = \mathbf{P}(t) - \mathbf{E}(s,t) - \mathbf{Q}(s,t) \qquad (14)$$

Here, $\mathbf{s}$ is the volume fraction of water within the pore volume of the soil, and $\theta$ represents soil moisture (the ratio of the volume of water to the unit volume of soil [$m^3\,m^{-3}$]), the subscripts $fc$ and $w$ represent field capacity and wilting point soil moisture, respectively. $t$ is time [s], $\mathbf{P}(t)$ is the hourly rate of precipitation representing water entering the model domain [m/h], $\mathbf{E}(s,t)$ is the hourly rate of evapotranspiration representing water leaving the model domain into the atmosphere [m/h], and $\mathbf{Q}(s,t)$ is the rate of deep drainage (vertical transport of water to deeper soil layers) and runoff (horizontal transport) [m/h]. $\theta_{fc}$ can be approximated as the porosity of soil, $\phi$. Given the semi-arid climate of our field site and considering that groundwater is largely disconnected from the control volume at this site (i.e., estimated groundwater depth > 75 m below the surface), we assume $\mathbf{Q}$ is zero so that no water leaves from the bottom of the bucket (at 20 m depth). This assumption, however, limits our ability to capture water loss due to lateral runoff and thus may lead to overestimation of water mass during storm seasons. We model $\mathbf{E}$ as the product of actual water saturation as measured by SMAP and potential evapotranspiration ($\mathbf{PET}$, m/h). Thus, after the above simplification, we arrive at the following ordinary differential equation:

$$\Delta z \cdot \phi \cdot \frac{d\mathbf{s}}{dt} = \mathbf{P}(t) - \mathbf{PET}(t) \cdot \frac{\mathbf{SMAP}(t)}{\phi}, \qquad (15)$$

We use the temperature-based Thornthwaite equation[68] to estimate $\mathbf{PET}$:

$$\mathbf{PET}(t) = 1.6 \left( \frac{10\mathbf{T}(t)}{I} \right)^a, \qquad (16)$$

where $a = 6.75 \cdot 10^{-7} \cdot I^3 - 7.71 \cdot 10^{-5} \cdot I^2 + 0.01792 \cdot I + 0.49239$, and the heat index $I = 83$ calculated from the temperature in summer. $\mathbf{T}(t)$ is the atmospheric temperature data (in degrees Celsius) from the NID station.

We use the Forward Euler method with a time step size of $\Delta t = 3$ hours, and the following parameters: $\Delta z = 20$ m, $\theta_{fc} \approx \phi = 0.25$ for the top 20 m depth[69], $\theta_w = 0.01$, to integrate the discretized version of Eq. (2):

$$\mathbf{s}_{n+1} = \mathbf{s}_n + \frac{\Delta t}{\Delta z \left( \theta_{fc} - \theta_w \right)} \left[ \mathbf{P}(t) - \mathbf{PET}(t) \frac{\mathbf{SMAP}(t)}{\phi} \right], \qquad (17)$$

where $\mathbf{s}_n$ represents saturation at the time step $n$. The choice of $\Delta z = 20$ m is based on the seismic data analysis that indicates the strongest sensitivity to meteorologic forcing at the highest frequency. The choice of $\phi = 0.25$, representing the average porosity over 20 m, takes into account the compaction effect at depth[69]. We find that initial saturation does not change the temporal trend of predicted moisture. The time step size of 3 hours is determined by the available data frequency for precipitation, SMAP, and temperature. Overall, the model is most truthful in its predicted temporal dynamics of moisture but bears

uncertainty in the amplitude of soil moisture, requiring future in situ field validation.

## Eddy-covariance measurements of evapotranspiration

We inferred a vadose zone water loss of 0.25 m/yr, which is consistent with USGS published eddy-covariance based evapotranspiration measurement of 0.2 m/yr at a sparse shrub site (hereafter refer to as shrub site) in the southwestern Mojave Desert[50]. Even though the evapotranspiration measurement (April 2018-March 2019) does not overlap with our observational period (July 2019–April 2022), the shrub site features vegetation coverage similar to that of our Ridgecrest filed site under comparable climate conditions (annual precipitation of 0.068 m/yr at shrub site vs. 0.05 m/yr in Ridgecrest)[50]. This similarity in vegetation coverage and climates assures similar estimations of subsurface water loss through both transpiration and evaporation processes. Future benchmarks would involve in situ field measurements and their incorporation into hydrological modeling to improve parameter estimation.

## Data availability

The cross-correlation product and multi-frequency dv/v generated in this study have been deposited in Zenodo under the accession code https://doi.org/10.5281/zenodo.12617908[70]. The precipitation data used in this study are available in the California Nevada River Forecast Center under accession code https://www.cnrfc.noaa.gov/arc_search.php. The groundwater well data used in this study are available in the Indian Wells Valley Groundwater Authority under accession code https://iwvgsp.com/. The surface temperature and surface soil moisture data used in this study are available from the National Snow and Ice Data Center under accession code https://nsidc.org/data/smap/data[71]. Source data are provided in this paper.

## Code availability

The GPU-based ambient noise cross-correlation code can be downloaded from https://github.com/zhichaoshen40/DAS_CC_GPU.git[72].

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

## Acknowledgements

We thank Santiago G. Solazzi for discussions on rock physics modeling from soil moisture to dv/v. This study is supported by the National Science Foundation CAREER #1848166 and the Resnick Institute of Sustainability. We are grateful to the field and technical support from Martin Karrrenbach, Lisa LaFlame, Vlad Bogdanov of Optasense Inc., Thomas Coleman of Silixa Inc., and Andrew Klesh of Jet Propulsion Laboratory. We thank the California Broadband Cooperative and JPL for providing access to the Digital 395 telecommunication fibers. Z.S. also thanks the support from the Weston Howland Jr. Postdoctoral Scholar Program at Woods Hole Oceanographic Institution. The research was partially carried out at the Jet Propulsion Laboratory, California Institute of Technology, under a contract with the National Aeronautics and Space Administration (80NM0018D0004).

## Author contributions

Z.S. and Z.Z. conceptualized this study. Z.S., Y.Y., X.F., and Z.Z. developed the methodology. Z.S. and Y.Y. processed the data, and conducted the investigations with X.F., K.H.A., E.B., and Z.Z. together. Z.S. and Y.Y. contributed to the visualization. E.B. and Y.Y. contributed to the data management. Z.S. and Y.Y. drafted the original manuscript, and all the authors reviewed, edited, and refined the manuscript.

## Competing interests

The authors declare no competing interests.
