## [Peer Review File · Nature Communications]

Fiber-optic seismic sensing of vadose zone soil moisture dynamicsREVIEWER COMMENTS

Reviewer #1 (Remarks to the Author):

Review of “Fiber seismic sensing of vadose zone soil moisture” by Zhichao Shen, Yan Yang, Xiaojing Fu, Kyra H. Adams, Ettore Biondi, and Zhongwen Zhan

The present document is a manuscript submitted by Shen et al. for publication in Nature Communications. It deals with a very innovative use of fiber optic sensing for hydrology through the technology often referred to as Distributed Acoustic Sensing (DAS). For the first time, the authors show that it is possible to use DAS technology to quantitatively monitor the dynamic of water content in the vadose zone

In my opinion, the authors used a sound and solid methodology to obtain their results, from state-of-the-art data processing and data correction (i.e., for temperature effects) to appropriate seismic and hydrological modeling. Their results are convincing and opens-up new and powerful ways to benefit from new or existing Fiber Optics for environmentally relevant studies of important societal problems, namely the monitoring of water content and drought in the critical zone. In my opinion, this works is a big step toward passive monitoring of hydrosystems by building on the recent advances of Fiber Optic. I do have minor comments to share (list below) but the paper is overall well written and easy to follow. Therefore, I strongly recommend its publication in Nature Communications.

Damien Jougnot

CNRS Senior Scientist

Sorbonne Université

List of minor comments:

Title: I suggest to clearly state “Fiber Optic” in the title and add the notion of dynamics as it is a major issue that the authors address in their work. It would give something like: “Fiber optic seismic sensing of vadose zone soil moisture dynamics”

Lines 62-69: To the best of my knowledge the passive seismic community has never done such study with an unprecedented lateral resolution. However, in their recent work, Gaubert-Bastide et al. (2022) did mention it as a potential step-up for hydro-seismology in

the near future. I suggest to cite this work in the introduction.

Figure 4: These results are fairly impressive and I was very enthusiastic as I was discovering them. However, I think that a simple flowchart describing the biggest steps of the modelling and inversion could help the future readers to understand it faster and/or reproduce the proposed approach. If such figure would be too big for the main paper, it could go into supplementary materials.

Reference:

Gaubert-Bastide, T., Garambois, S., Bordes, C., Voisin, C., Oxarango, L., Brito, D., & Roux, P. (2022). High-resolution monitoring of controlled water table variations from dense seismic-noise acquisitions. *Water Resources Research*, 58(8), e2021WR030680.

Reviewer #2 (Remarks to the Author):

This is an interesting paper about applying passive seismic interferometry to characterize soil moisture fluctuation. Instead of calling it a technical paper, I would like to refer to it as a case study of the soil moisture problem.

I do not find any major flaw in this work, probably because I'm not a soil geoscientist by training; I can only comment on its technical merit from the seismological point of view.

Considering the wide application of fiber optical sensing technologies, the technical advance of this paper is weak. Obviously, there are no compelling imaging/inversion/analysis advances from the paper, except for another application of fiber optics in passive seismic interferometry. The passive seismic interferometry technique is emerging, likely because of the recent work by Mao et al. 2022 (NatureComm). Here, the only difference is it is applied to a different sensor, DAS, which, in my opinion, is not a ground-breaking advance but rather just a finer-scale case study. The resolution advantage of DAS is widely known, and there this advantage is underscored again in a new problem, which is no wonder.

One improvement can be to benchmark the resulting soil moisture characteristics with a state-of-the-art approach. A larger-scale scenario can be benchmarked with the InSAR

surface deformation change, as reported in Mao 2022, but I'm not sure what the best reference approach is for the finer-scale soil moisture scenario. A good idea might be to get samples from the field as the ground truth, which corroborates what is obtained through DAS observation.

A shortcoming is that there is little discussion on potential pitfalls when applying the fiber optics approach for conducting such characterization. DAS is not a panacea, and there must be many caveats when this technology is deployed in practice. More discussions and numerical tests on various processing strategies and parameterization could change the results, which is strongly needed.

The data and code links provided in the manuscript are all end products, which are not helpful in reproducing and verifying the processing and analysis workflow introduced in the paper. The authors should provide the major processing workflow, as well as raw or intermediate data products so that the readers have sufficient tools and datasets to reproduce and verify the results presented in this paper.

Reviewer #3 (Remarks to the Author):

Review of "Fiber seismic sensing of vadose zone soil moisture" by Shen et al, 2024.

Review done by Marine Denolle [email address redacted]

Summary:

The manuscript presents fascinating results of using cutting-edge technology (distributed acoustic sensing) with a sophisticated model of soil moisture changes. The work builds upon past literature from the seismological community to infer subsurface hydrological properties using ambient field analysis, with Illien et al., 2021 taking an example of the most recent sophisticated work on the topic. This manuscript has the added value that the soil model is quite sophisticated and fits well with the long-term trend in this 2-year survey. The amount of data processed is quite impressive, in my opinion.

The study area is in Ridgecrest. The array was deployed as an opportunity for post-earthquake response and coincidentally is at the edge of an interesting drying lake (China Lake). The strength of DAS is the ability for distributed sensing along a fiber. The team could leverage this a little bit more in this study, maybe by incorporating their cool Supp. Fig 6 in the main manuscript, for instance. There could also be improvements in using their fantastic shear wave models for spatial variations in the sensitivity of the waves and depth. I feel like discussing the spatial heterogeneity of their finding would strengthen the use of DAS for soil moisture sensing. In fact, the major drawback of remote-sensing soil measurements is not just the lack of depth sensitivity or the depth-integrated data product from GRACE but rather the large spatial footprint of these measurements. The team here can potentially address the spatial heterogeneity of soil moisture in the context of the arid basins.

Main comments:

- The hydrological model first pretty well the data and it seems quite sophisticated. It is great to finally see more hydrology in a seismo-hydrology paper! I am still a bit confused on what is being fit, a simple equation that says $dv/v(x,t) = f(\text{volumetric water change}, x, t)$ to really state what is data, what is model, would be helpful. I got lost in the derivation of elastic moduli and dv/v .
- It would have been quite simpler to directly do a time-lapse imaging with ballistic waves. S

Detailed Comments.

L17: "large spatiotemporal scales" -> isn't it small spatial scales and short-term that there is a problem? Satellite measurements are large-scale.

L22: "Our observations [...] independently validated with a zero-dimensional hydrological model." -> Isn't it that the model is validated by the observations?

L24: evapotranspiration -> is there vegetation in the desert? I thought evaporation would be sufficient.

L26: "drought risk" could be placed under the regional context. Globally, flooding is also a concern of climate change.

L48: “time domain reflectometer [...] operational cost of long-term monitoring is often prohibitive.” you may also refer to the GNSS-derived soil moisture sensors. See a review by Edokossi et al. (2020) and Larson et al. (2010) for some of the earliest works that are free by-products of GNSS operations.

L141: The magnitude of the dv/v is mostly a matter of choice in seismic frequency and in the inter-station distance. Illien et al. (2021) found greater changes in the 4-8Hz using single-station measurements than your study finds. Clements and Denolle (2023) use single station correlation, 2-4Hz, and find the same order of magnitude as your study. Laudi et al. (2023) also find the same order of magnitude.

Evapotranspiration: I am not a hydrologist, but I wonder if simply stating “evaporation” would not suffice because the study takes place in the desert, and I would anticipate more evaporation than plant transpiration.

L168: “soil moisture dynamics” -> “soil moisture budget”? There is no modeling of groundwater flow but rather a constitutive modeling of elastic properties.

Equation 16: does PET take a subscript i ?

L200 “in contradiction to the hydrological model (Fig. 4).” There is a lot of variability in the dv/v time series, so I would not say that it contradicts much. In fact, I am not sure that the hydrological model can assume constant properties. Groundwater flow in partially saturated soils seems a bit more complicated (Goncalves et al., 2020, for example).

Figure 4: There is something interesting at 4km. It appears at a low velocity in the shear wave velocity model, and it appears to have an increased sensitivity to thermoelastic stresses. Does it coincide with the channel that is visible in the Google Earth images (HWY 395 business and N Brady St intersection)

Methods:

- Why do dv/v when directly picking the surface wave? It would be more “deterministic” to do the travel time measurement using simple cross-correlation and over various frequencies to get a dispersion curve, then invert and get the absolute velocity change.
- The increase of dv/v with frequency is also a characteristic of the exponentially decaying sensitivity of the waves with depth. You may refer to Obermann et al. (2013); 2016 for the body waves and coda examples, and in Yuan et al. (2021), there is a similar case for the surface waves (early coda or late ballistic). Mordret et al. (2020) is an interesting approach for arrays like DAS.
- It seems a bit of a missed opportunity not to do the depth inversion, given that there is a very nice velocity model. That said, the study is still excellent!
- While I am not a hydrologist, I find the sophistication of the model for soil moisture excellent and a leap forward in the field of environmental seismology.
- Inversion of dv/v . Could you summarize in the method, section 2, what is inverted. I understand how to get from soil type to bulk and shear moduli, then to shear wavespeed. Soil moisture is also predicted from precipitation, temp, and SMAP. What is the relation that predicts water loss from dv/v ?

Data and code availability:

The team mentioned a GPU-enabled cross correlation workflow, they could/should share their codes. I also recommend writing the output file as a CSV instead of MAT since the data output is a table (time, space, dv/v) to be more standard with community use.

References:

K. M. Larson, J. J. Braun, E. E. Small, V. U. Zavorotny, E. D. Gutmann and A. L. Bilich, "GPS Multipath and Its Relation to Near-Surface Soil Moisture Content," in IEEE Journal of Selected Topics in Applied Earth Observations and Remote Sensing, vol. 3, no. 1, pp. 91-99, March 2010, doi: 10.1109/JSTARS.2009.2033612. keywords: {Global Positioning System;Satellites;Soil moisture;Signal to noise ratio;Receivers;Antennas;Global positioning system;remote sensing;soil measurements},

Gonçalves, R.D., Teramoto, E.H., Engelbrecht, B.Z., Alfaro Soto, M.A., Chang, H.K. and van Genuchten, M.T. (2020), Quasi-Saturated Layer: Implications for Estimating Recharge and Groundwater Modeling. *Groundwater*, 58: 432-440. <https://doi.org/10.1111/gwat.12916>

Yuan C, Bryan J, Denolle M. Numerical comparison of time-, frequency-and wavelet-domain methods for coda wave interferometry. *Geophysical Journal International*. 2021 Aug;226(2):828-46.

Edokossi K, Calabia A, Jin S, Molina I. GNSS-reflectometry and remote sensing of soil moisture: A review of measurement techniques, methods, and applications. *Remote Sensing*. 2020 Feb 12;12(4):614.

Larson KM, Braun JJ, Small EE, Zavorotny VU, Gutmann ED, Bilich AL. GPS multipath and its relation to near-surface soil moisture content. *IEEE Journal of Selected Topics in Applied Earth Observations and Remote Sensing*. 2009 Nov 10;3(1):91-9.

Edokossi, Komi, Andres Calabia, Shuanggen Jin, and Iñigo Molina. 2020. "GNSS-Reflectometry and Remote Sensing of Soil Moisture: A Review of Measurement Techniques, Methods, and Applications" *Remote Sensing* 12, no. 4: 614. <https://doi.org/10.3390/rs12040614>

Illien L, Andermann C, Sens-Schönfelder C, Cook KL, Baidya KP, Adhikari LB, Hovius N. Subsurface moisture regulates Himalayan groundwater storage and discharge. *AGU Advances*. 2021 Jun;2(2):e2021AV000398.

Obermann A, Planès T, Larose E, Campillo M. Imaging preeruptive and coeruptive structural and mechanical changes of a volcano with ambient seismic noise. *Journal of Geophysical Research: Solid Earth*. 2013 Dec;118(12):6285-94.

Obermann A, Planès T, Hadziioannou C, Campillo M. Lapse-time-dependent coda-wave depth sensitivity to local velocity perturbations in 3-D heterogeneous elastic media. *Geophysical Journal International*. 2016 Oct 1;207(1):59-66

Mordret A, Courbis R, Brenguier F, Chmiel M, Garambois S, Mao S, Boué P, Campman X, Lecocq T, Van der Veen W, Hollis D. Noise-based ballistic wave passive seismic monitoring– Part 2: surface waves. *Geophysical Journal International*. 2020 Apr;221(1):692-705.

We thank all the reviewers for providing extremely valuable comments on our manuscript, based on which we revised the manuscript. The following are detailed replies (in purple) to comments (in black). The sentences modified in the manuscript are shown at the end of each corresponding reply and marked as italic text colored in blue. The line number (starting with “L”) in this response letter refers to the line number in the tracked version of the main text.

Reviewer #1 (Dr. Damien Jougnot):

The present document is a manuscript submitted by Shen et al. for publication in Nature Communications. It deals with a very innovative use of fiber optic sensing for hydrology through the technology often referred to as Distributed Acoustic Sensing (DAS). For the first time, the authors show that it is possible to use DAS technology to quantitatively monitor the dynamic of water content in the vadose zone.

In my opinion, the authors used a sound and solid methodology to obtain their results, from state-of-the-art data processing and data correction (i.e., for temperature effects) to appropriate seismic and hydrological modeling. Their results are convincing and opens-up new and powerful ways to benefit from new or existing Fiber Optics for environmentally relevant studies of important societal problems, namely the monitoring of water content and drought in the critical zone. In my opinion, this work is a big step toward passive monitoring of hydrosystems by building on the recent advances of Fiber Optic. I do have minor comments to share (list below) but the paper is overall well written and easy to follow. Therefore, I strongly recommend its publication in Nature Communications.

Reply: Thank you for your encouraging comments. We have diligently revised the manuscript in accordance with your comments. Please find the point-to-point response below.

List of minor comments:

Title: I suggest to clearly state “Fiber Optic” in the title and add the notion of dynamics as it is a major issue that the authors address in their work. It would give something like: “Fiber optic seismic sensing of vadose zone soil moisture dynamics”.

Reply: Thank you for the suggestion. We revised the title accordingly to “*Fiber-optic seismic sensing of vadose zone soil moisture dynamics*” and replaced the “fiber seismic sensing” with “fiber-optic seismic sensing” accordingly in the text.

Lines 62-69: To the best of my knowledge the passive seismic community has never done such study with an unprecedented lateral resolution. However, in their recent work, Gaubert-Bastide et al. (2022) did mention it as a potential step-up for hydro-seismology in the near future. I suggest to cite this work in the introduction.

Reference,

Gaubert-Bastide, T., Garambois, S., Bordes, C., Voisin, C., Oxarango, L., Brito, D., & Roux, P. (2022). High-resolution monitoring of controlled water table variations from dense seismic-noise acquisitions. Water Resources Research, 58(8), e2021WR030680.

Reply: Thank you for the suggestion. We added the reference accordingly in our introduction.

L54-55

Recently, time-lapse seismology has shown great promise to characterize subsurface hydrological processes using the seismic velocity change (dv/v) as an indicator of water saturation²⁴⁻²⁷.

Figure 4: These results are fairly impressive and I was very enthusiastic as I was discovering them. However, I think that a simple flowchart describing the biggest steps of the modelling and inversion could help the future readers to understand it faster and/or reproduce the proposed approach. If such figure would be too big for the main paper, it could go into supplementary materials.

Reply: Thank you very much for the suggestion. We added a Supplementary Fig. 4 to describe the major steps in our inversion and modeling to improve the clarity and referenced it properly in the main text and Methods. As suggested by Reviewer#3, we also included an explanatory sentence with simplified equation in the beginning of Methods Section 3.

L314-316

Here we briefly summarize the Solazzi model that enables a direct prediction of dv/v from a given vadose zone soil moisture change, which follows $dv/v(t) = f(\text{vadose zone soil moisture change, lithology model, } t)$ (Supplementary Fig. 4a).

Reviewer #2:

This is an interesting paper about applying passive seismic interferometry to characterize soil moisture fluctuation. Instead of calling it a technical paper, I would like to refer to it as a case study of the soil moisture problem.

I do not find any major flaw in this work, probably because I'm not a soil geoscientist by training; I can only comment on its technical merit from the seismological point of view.

Reply: Thank you for finding our paper interesting and noting that there are no major flaws in our work.

Considering the wide application of fiber optical sensing technologies, the technical advance of this paper is weak. Obviously, there are no compelling imaging/inversion/analysis advances from the paper, except for another application of fiber optics in passive seismic interferometry. The passive seismic interferometry technique is emerging, likely because of the recent work by Mao et al. 2022 (NatureComm). Here, the only difference is it is applied to a different sensor, DAS, which, in my opinion, is not a ground-breaking advance but rather just a finer-scale case study. The resolution advantage of DAS is widely known, and there this advantage is underscored again in a new problem, which is no wonder.

Reply: Thank you for the comment. Indeed, we agree that the time-lapse seismic interferometry is not new in seismology and the high-resolution capacity of DAS has also been widely recognized in recent years. We'd like to clarify that our objective is to strengthen and advance the theoretical framework between environmental seismology and (vadose zone) hydrology, instead of advancing the ambient noise method. We hope that our study demonstrates that fiber-optic seismic sensing provides a new means to measure vadose zone water dynamics in a high-resolution manner, which is challenging for current remote sensing techniques.

Therefore, we'd like to highlight two points for our study. The first one is that we demonstrate the viability of passive seismic sensing for sustained high-resolution hydrology monitoring of the shallow subsurface (i.e., vadose zone water), given the massive data volume processed. We note that this study is different from Mao et al. (2022), which focuses on deeper hydrological processes related to groundwater, due to finer channel spacing and higher frequencies. The 2.5-year DAS raw data yields a total volume of 80 TB, much larger than the data size of previous dv/v studies. Our algorithm ensures an efficient dv/v observation for handling large data volume. We added a sentence in Methods Section 1.1 to state the data size.

The second technical advancement is that our fiber-optic seismic sensing combines seismic sensing and classic rock physics. This enables us to map real seismic data to vadose zone soil moisture changes and compare it to 0D hydrological modeling coupled with meteorological data and SMAP data. We have added a Supp. Fig. 4 and revised relevant sentences to highlight the sophistication and advancement of our approach. We refer to our replies to the last comment by Reviwer#1 and to the first main comment by Reviewer#3 for details.

L248-249

The 2.5-year continuous raw data recorded by our Ridgecrest DAS array yields a total volume of 80 TB for ambient noise cross-correlations.

One improvement can be to benchmark the resulting soil moisture characteristics with a state-of-the-art approach. A larger-scale scenario can be benchmarked with the InSAR surface deformation change, as reported in Mao 2022, but I'm not sure what the best reference approach is for the finer-

scale soil moisture scenario. A good idea might be to get samples from the field as the ground truth, which corroborates what is obtained through DAS observation.

Reply: Thank you for the suggestion. Large-scale remote sensing can hardly observe soil moisture changes at depths, as we summarized in our introduction. It is a fair point to consider using in-situ field measurements as a direct benchmark for the dv/v observations. Unfortunately, we lack in-situ samples during the observational period. Alternatively, our validation approach is to compare dv/v-inferred soil moisture change to nearby eddy-covariance measurement under similar vegetation coverage and climate condition. The consistency between them suggests that our approach and estimates are reasonable. We added two sentences in Methods Section 5 to clarify the environmental similarity between our field site and nearby shrub site for eddy-covariance measurement, and the incorporation of in-site field samples for future benchmarks.

L429-433

This similarity in vegetation coverage and climates assures similar estimations of subsurface water loss through both transpiration and evaporation processes. Future benchmarks would involve in-situ field measurements and their incorporation into hydrological modeling to improve parameter estimation.

A shortcoming is that there is little discussion on potential pitfalls when applying the fiber optics approach for conducting such characterization. DAS is not a panacea, and there must be many caveats when this technology is deployed in practice. More discussions and numerical tests on various processing strategies and parameterization could change the results, which is strongly needed.

Reply: Thank you for the suggestion. Indeed, the success of applying fiber-optic sensing for vadose zone soil moisture sensing depends on a set of factors, including the noise source distribution and cable coupling. In urban areas, noise sources are mostly from anthropological activities. Poor cable coupling would also contribute to reduced signal amplitudes. For our Ridgecrest DAS array, we do benefit from the fiber-optic cable inside the conduits offering excellent coupling and the straight array along the major highway for stationary-phase zone stacking. In addition, variations in subsurface lithology can also translate uncertainties into the magnitude of our inferred soil moisture changes. Thus, lithology calibrations for different soils are needed for accurate amplitude estimates in soil moisture changes. Nonetheless, it is also worth noting that the temporal trend of resolved soil moisture changes remains independent of subsurface lithology. Overall, we'd like to clarify that our paper is to showcase that fiber-optic seismic sensing provides a new means to probe vadose zone soil moistures in a high-resolution manner. We added a few sentences in the main text to highlight potential pitfalls in practical applications.

L231-237

With the Ridgecrest DAS array, we demonstrate the viability of fiber-optic seismic sensing for high-resolution vadose zone soil moisture dynamics. For practical applications of fiber-optic seismic sensing in various environments, the noise source distribution, poor cable coupling, and unclear subsurface lithology would lead to uncertainties in the fiber-optically estimated vadose zone water content changes. Combined with continued seismic sensing, future work involving direct in-situ measurements for different soil types and regional scale modeling will increase the robustness of the inverted vadose zone soil moisture dynamics.

The data and code links provided in the manuscript are all end products, which are not helpful in reproducing and verifying the processing and analysis workflow introduced in the paper. The

authors should provide the major processing workflow, as well as raw or intermediate data products so that the readers have sufficient tools and datasets to reproduce and verify the results presented in this paper.

Reply: Thank you for the comment. It is challenging to share the raw 80 TB data. We opened our intermediate product (i.e., CC used for dv/v measurements) to ensure readers can reproduce and verify our results. We revised the data section accordingly.

L436-446

The cross-correlation product and multi-frequency dv/v map produced by this study is available at [https://data.caltech.edu/records/g4hbc-](https://data.caltech.edu/records/g4hbc-0mt45?token=eyJhbGciOiJIUzUxMiJ9.eyJpZCI6Ijg1MGFiNmI0LWNjNmEtNDgzMCIhM2Q2LTgwMTUzZWl5NDBhNyIsImRhdGEiOiJ9LCJyYW5kb20iOiJzNTRkZGI1ODU5MjBhOTkxZjE3MDY2MWE2YWQ5MjF1NiJ9.dsJropZBGFAPLi8VqhXLZfttg_x6BqTixHtV9UJ-QJIsQ-AWOUmxHqPYVH9RffJsPuz4xjsTKNMrw1H5RUf2Q)

[0mt45?token=eyJhbGciOiJIUzUxMiJ9.eyJpZCI6Ijg1MGFiNmI0LWNjNmEtNDgzMCIhM2Q2LTgwMTUzZWl5NDBhNyIsImRhdGEiOiJ9LCJyYW5kb20iOiJzNTRkZGI1ODU5MjBhOTkxZjE3MDY2MWE2YWQ5MjF1NiJ9.dsJropZBGFAPLi8VqhXLZfttg_x6BqTixHtV9UJ-QJIsQ-AWOUmxHqPYVH9RffJsPuz4xjsTKNMrw1H5RUf2Q](https://data.caltech.edu/records/g4hbc-0mt45?token=eyJhbGciOiJIUzUxMiJ9.eyJpZCI6Ijg1MGFiNmI0LWNjNmEtNDgzMCIhM2Q2LTgwMTUzZWl5NDBhNyIsImRhdGEiOiJ9LCJyYW5kb20iOiJzNTRkZGI1ODU5MjBhOTkxZjE3MDY2MWE2YWQ5MjF1NiJ9.dsJropZBGFAPLi8VqhXLZfttg_x6BqTixHtV9UJ-QJIsQ-AWOUmxHqPYVH9RffJsPuz4xjsTKNMrw1H5RUf2Q) (restricted access for review process, but will be publicly available at the time of publication).

Reviewer #3 (Dr. Marine Denolle):

The manuscript presents fascinating results of using cutting-edge technology (distributed acoustic sensing) with a sophisticated model of soil moisture changes. The work builds upon past literature from the seismological community to infer subsurface hydrological properties using ambient field analysis, with Illien et al., 2021 taking an example of the most recent sophisticated work on the topic. This manuscript has the added value that the soil model is quite sophisticated and fits well with the long-term trend in this 2-year survey. The amount of data processed is quite impressive, in my opinion.

The study area is in Ridgecrest. The array was deployed as an opportunity for post-earthquake response and coincidentally is at the edge of an interesting drying lake (China Lake). The strength of DAS is the ability for distributed sensing along a fiber. The team could leverage this a little bit more in this study, maybe by incorporating their cool Supp. Fig 6 in the main manuscript, for instance. There could also be improvements in using their fantastic shear wave models for spatial variations in the sensitivity of the waves and depth. I feel like discussing the spatial heterogeneity of their finding would strengthen the use of DAS for soil moisture sensing. In fact, the major drawback of remote-sensing soil measurements is not just the lack of depth sensitivity or the depth-integrated data product from GRACE but rather the large spatial footprint of these measurements. The team here can potentially address the spatial heterogeneity of soil moisture in the context of the arid basins.

Reply: Thank you for your encouraging comments. We moved the content of Supp. Fig. 6 to Fig. 4b but presenting it in a clearer version of later variations of inverted total soil moisture loss. Accordingly, we added a few sentences in the main text to discuss the spatial variability and a short paragraph in Methods Section 3.2 to describe the calculation.

L211-218

Furthermore, leveraging our dv/v data across the entire cable, we invert the lateral variations in total soil moisture loss over the 2.5-year observational period (see Methods for details; Fig. 4b). Despite variations in the assumed lithology models, the eastern end (i.e., 6-8 km) appears to experience a greater loss in vadose zone soil moisture compared to the western end (Fig. 4b), suggesting a high sensitivity of evapotranspiration rate to localized environmental forcing. Achieving such high-resolution spatial variations is challenging with satellite-based remote sensing techniques, but feasible with our fiber-optic seismic sensing.

L371-382

Repeating the inversion of daily volumetric water content changes for all DAS channels, we can estimate the total vadose zone soil moisture loss across the 8-km cable over the 2.5 years of our observational period. To ensure robust estimates, we apply the following two criteria for data quality control: (a). For a given channel and a specific date, the square root of the resolved mean square misfit must be less than 0.1%; (b). For a given channel, the number of inverted daily soil moisture changes must exceed 70% of the total days of our observation period to robustly determine total soil moisture loss. To account for lithology model uncertainties, we use the two end-member models as estimate bounds to represent the variability of the total soil moisture loss. The resulting lateral variations of total vadose zone soil moisture loss are shown in Fig. 4b, with two gaps at 2-3 km and round 4-km mark. The gaps are due to insufficient high-quality data that can meet our criteria. For example, the dv/v uncertainty is generally large around the 4-km mark (Supplementary Fig. 7).

We also thank you for noting our high-resolution surface wave tomography. Indeed, it achieves high-resolution at top 150 m, while its uncertainty is relatively high when zooming into the top 20

m. We refer to our reply to your second main comment for more details. We have also diligently revised the manuscript in accordance with your comments listed below. Please find the point-to-point response below.

Main comments:

- The hydrological model fits pretty well the data and it seems quite sophisticated. It is great to finally see more hydrology in a seismo-hydrology paper! I am still a bit confused on what is being fit, a simple equation that says $dv/v(x,t) = f(\text{volumetric water change}, x, t)$ to really state what is data, what is model, would be helpful. I got lost in the derivation of elastic moduli and dv/v .

Reply: Thank you very much for the suggestion. We added a few sentences in the main text and Methods Section 3.1 to clarify what variables are used and what are modelled. Also, as suggested by Reviewer#1, we included a flowchart in Supp. Fig. 4 and cited it properly in the manuscript.

L179-182

The forward model calculates dv/v as a function of soil moisture profile and a given lithology model, which enables us to invert for water content changes in the top 20 m that can best fit dv/v measurements across all frequencies (see Methods for details; Supplementary Figs. 4a and 5).

L200-202

For independent validation, we further use a zero-dimensional hydrological model forced with meteorological and SMAP data products to simulate subsurface volumetric water content changes at our field site^{49,50} (Fig. 1a; see Methods for details; Supplementary Fig. 4b).

L314-316

Here we briefly summarize the Solazzi model, that enables a direct prediction of dv/v from a given vadose zone soil moisture change, which follows $dv/v(t) = f(\text{vadose zone soil moisture change, lithology model, } t)$ (Supplementary Fig. 4a).

- It would have been quite simpler to directly do a time-lapse imaging with ballistic waves.

Reply: Thank you for the comment. Our tomography model using ballistic surface waves indeed achieves high depth resolution in the top 150 m. But its uncertainty is relatively high when zooming into very shallow depths, hindering us to further conduct a time-lapse imaging, as the uncertainty may surpass the temporal changes in seismic velocity. Nonetheless, our observations pinpoint dv/v sources to the shallow vadose zone and constrain the volumetric water content change over top 20 m. We added two sentences in Methods Section 1.2 to clarify.

L281-284

Although our shear wave velocity model achieves high resolution in the top 150 m, the velocity uncertainty at very shallow depths remains high³². Future attempts incorporating high-quality higher-frequency ballistic waves would enhance the depth resolution of our tomography model.

Detailed Comments.

L17: “large spatiotemporal scales” -> isn’t it small spatial scales and short-term that there is a problem? Satellite measurements are large-scale.

Reply: Indeed, short-term fine-scale observations provide valuable insights for vadose zone soil moisture dynamics. The use of “larger spatiotemporal scales” here is to highlight the scalable and long-term cost-effective nature of fiber-optic sensing. We added “high-resolution” in the sentence to emphasize the short-term, fine-scale observations over large spatiotemporal scales.

L17-18:

However, the inability to observe high-resolution vadose zone soil moisture dynamics over large spatiotemporal scales hinders quantitative characterization.

L22: “Our observations [...] independently validated with a zero-dimensional hydrological model.”
-> Isn't it that the model is validated by the observations?

Reply: Indeed, observations also validate our hydrological modeling, given their consistency. In fact, both models are quantitatively validated by the nearby eddy-covariance based measurements. We replaced “independently validated” with “consistent” to convey our meaning more precisely.

L20-24:

Our observations in Ridgecrest, California reveal sub-seasonal precipitation replenishments and a prolonged drought in the vadose zone, consistent with a zero-dimensional hydrological model. Our results suggest a significant water loss of 0.25 m/year through evapotranspiration at our field side, validated by eddy-covariance based measurements in nearby region.

L24: evapotranspiration -> is there vegetation in the desert? I thought evaporation would be sufficient.

Reply: Yes, Ridgecrest is sparsely covered by shrubs (Figure R1), which can take water away from the shallow subsurface through transpiration. We improved Methods Section 5 to further clarify the similarity in vegetation coverage between Ridgecrest and the shrub site for eddy-covariance based measurements.

L425-431

Even though the evapotranspiration measurement (April 2018-March 2019) does not overlap with our observational period (July 2019-April 2022), the shrub site features vegetation coverage similar to that of our Ridgecrest field site under comparable climate conditions (annual precipitation of 0.068 m/yr at shrub site vs. 0.05 m/yr in Ridgecrest)⁵¹. This similarity in vegetation coverage and climates assures similar estimations of subsurface water loss through both transpiration and evaporation processes.

Figure R1. Street view of Ridgecrest DAS array from Google Maps, facing west.

L26: “drought risk” could be placed under the regional context. Globally, flooding is also a concern of climate change.

Reply: Thanks for the suggestion. We specified “regional” before “drought risk” in the abstract and main text to be more accurate.

L27 & L237

Given the escalated regional drought risk under climate change.

L48: “time domain reflectometer [...] operational cost of long-term monitoring is often prohibitive.” you may also refer to the GNSS-derived soil moisture sensors. See a review by Edokossi et al. (2020) and Larson et al. (2010) for some of the earliest works that are free by-products of GNSS operations.

Reference:

Larson KM, Braun JJ, Small EE, Zavorotny VU, Gutmann ED, Bilich AL. GPS multipath and its relation to near-surface soil moisture content. *IEEE Journal of Selected Topics in Applied Earth Observations and Remote Sensing*. 2009 Nov 10;3(1):91-9.

Edokossi, Komi, Andres Calabia, Shuanggen Jin, and Iñigo Molina. 2020. "GNSS-Reflectometry and Remote Sensing of Soil Moisture: A Review of Measurement Techniques, Methods, and Applications" *Remote Sensing* 12, no. 4: 614. <https://doi.org/10.3390/rs12040614>

Reply: Indeed, the emerging GNSS based techniques provide a low-cost means to measure surface soil moisture content but are sensitive to a depth of centimeters. To avoid confusion, we mentioned the GNSS based techniques in the sentence of remote sensing and cited the references properly.

L40-42

Modern microwave remote sensing missions, such as the SMAP¹² (Soil Moisture Active Passive) and SMOS¹³ (Soil Moisture and Ocean Salinity), and GNSS (Global Navigation Satellite System) based techniques^{14,15} provide good estimates of global soil moisture only down to ~5 centimeters at 10-40 km spatial resolution every few days and can further extend to the 1-m root zone using data assimilation¹⁶.

L141: The magnitude of the dv/v is mostly a matter of choice in seismic frequency and in the inter-station distance. Illien et al. (2021) found greater changes in the 4-8Hz using single-station measurements than your study finds. Clements and Denolle (2023) use single station correlation, 2-4Hz, and find the same order of magnitude as your study. Laudi et al. (2023) also find the same order of magnitude.

Reference:

Illien L, Andermann C, Sens-Schönfelder C, Cook KL, Baidya KP, Adhikari LB, Hovius N. Subsurface moisture regulates Himalayan groundwater storage and discharge. *AGU Advances*. 2021 Jun;2(2):e2021AV000398.

Laudi, L., Agius, M. R., Galea, P., D'Amico, S. & Schimmel, M. Monitoring of Groundwater in a Limestone Island Aquifer Using Ambient Seismic Noise. *Water* 15, 2523 (2023).

Clements, T. & Denolle, M. A. The Seismic Signature of California's Earthquakes, Droughts, and Floods. *Journal of Geophysical Research: Solid Earth* 128, e2022JB025553 (2023).

Reply: Thank you for the suggestion. We added two sentences to clarify this point.

L145-149

Previous studies attribute seasonality in long-period seismic signals to groundwater level changes, but their reported dv/v amplitudes are one order of magnitude smaller than this study^{25,26}. Such amplitude discrepancy arises from the choice of frequency band and inter-station distance. In fact,

previous studies using high-frequency seismic waves yield the same order of magnitude variation in dv/v amplitudes^{33–35}.

Evapotranspiration: I am not a hydrologist, but I wonder if simply stating “evaporation” would not suffice because the study takes place in the desert, and I would anticipate more evaporation than plant transpiration.

Reply: Thank you for the comment. Evapotranspiration is the correct term in hydrology. We also observe that the Ridgecrest DAS array is sparsely covered by vegetations (Figure R1), which could take water away from the shallow subsurface through transpiration. We added a sentence in Methods Section 5 to clarify it.

L429-431

This similarity in vegetation coverage and climates assures similar estimations of subsurface water loss through both transpiration and evaporation processes.

L168: “soil moisture dynamics” -> “soil moisture budget”? There is no modeling of groundwater flow but rather a constitutive modeling of elastic properties.

Reply: Thank you for the suggestion. In fact, our inverted soil moisture change in Figure 4a is a time series while “budget” typically lacks a time dimension. To avoid confusion, we added a phrase after “dynamics” to clarify this.

L177

To further quantify the soil moisture dynamics (i.e., time series of soil moisture budget) using dv/v observations, ...

Equation 16: does PET take a subscript i?

Reply: We corrected it accordingly.

Eq. 16

$$PET(t) = 1.6 \left(\frac{10T(t)}{I} \right)^a$$

L200 “in contradiction to the hydrological model (Fig. 4).” There is a lot of variability in the dv/v time series, so I would not say that it contradicts much. In fact, I am not sure that the hydrological model can assume constant properties. Groundwater flow in partially saturated soils seems a bit more complicated (Goncalves et al., 2020, for example).

Reference:

Gonçalves, R.D., Teramoto, E.H., Engelbrecht, B.Z., Alfaro Soto, M.A., Chang, H.K. and van Genuchten, M.T. (2020), *Quasi-Saturated Layer: Implications for Estimating Recharge and Groundwater Modeling*. *Groundwater*, 58: 432-440. <https://doi.org/10.1111/gwat.12916>

Reply: Thank you for the suggestion. We weakened the tone using “differing from” and added a sentence to state the simplification of our model which does not consider surface runoff.

L222-226

Unlike the rapid replenishment from the two rainfall sequences in 2020, our inverted declining trend of soil moisture is not perturbed by the precipitation sequence in December 2021, differing from the hydrological model (Fig. 4a). This discrepancy may arise from oversimplification of the hydrological model, which does not consider surface runoff.

Figure 4: There is something interesting at 4km. It appears at a low velocity in the shear wave velocity model, and it appears to have an increased sensitivity to thermoelastic stresses. Does it

coincide with the channel that is visible in the Google Earth images (HWY 395 business and N Brady St intersection)

Reply: Thank you for noting the tomographic and geographic feature around 4 km. The channel near the intersection of HWY395 and N Brady St is located at the 5-km mark of our Ridgecrest DAS array. The cable at 4 km corresponds to the intersection between HWY395 and Micah St, where we do not observe distinct geographic features. In addition, the inverted soil moisture content curve in Fig. 4 is derived from the cable section near 7.2 km, which does not use the data around the 4-km mark.

In fact, we do observe clear scattered surface waves around 4 km in the cross-correlations (Supp. Fig. 5), which has been used by *Yang et al. (2023)* to confirm a geologically inferred fault and constrain its properties. The scattered waves do affect the quality of our dv/v measurements, leading to larger uncertainties of dv/v measurements (vertical stripes in Supp. Fig. 7) and a lateral data gap in our new Fig. 4b. We added a supplementary figure showing the dv/v uncertainty and included a few sentences in Methods to clarify this point.

L254-255

We also observe clear scattered surface waves at ~4 km, which has been suggested as the seismic signatures of a geologically inferred fault⁵³.

L263-266

Nonetheless, notable large dv/v uncertainties are observed near the 4-km mark along the cable (vertical stripes in Supplementary Fig. 7), which is attributed to scattered surface waves affecting the signal quality of retrieved direct surface waves (Supplementary Fig. 6).

L380-382

The gaps are due to insufficient high-quality data that can meet our criteria. For example, the dv/v uncertainty is generally large around the 4-km mark (Supplementary Fig. 7).

Methods:

- Why do dv/v when directly picking the surface wave? It would be more “deterministic” to do the travel time measurement using simple cross-correlation and over various frequencies to get a dispersion curve, then invert and get the absolute velocity change.

Reply: The selection of the time-domain cross-correlation method and the frequency-domain cross-spectrum method is essentially a choice of preferred weighting approach within a frequency band. For example, if the signal spectrum at a certain frequency is significantly higher than those of other frequencies, the time-domain cross-correlation tends to put more weighting in that frequency, while the cross-spectrum method could take an average dt measurements over the frequencies within the bandwidth. Given our high-quality cross-correlations, both choices should be basically equivalent. Here we use the cross-spectrum method, due to its wide use in dv/v studies and effective uncertainty quantification. We added a Supp. Fig. 7 showing the resulting dv/v uncertainties and a few sentences in Methods to improve the clarity.

L260-263

Here, we use the cross-spectrum method for its extensive use and effective uncertainty quantification^{25,57}. Benefitting from our high-quality cross-correlations, the resulting dv/v uncertainties are generally small, with the majority falling below 0.04% (Supplementary Fig. 7).

L266-269

Other methods, such as a straightforward time-domain cross-correlation and wavelet-based approaches, have also demonstrated great performance in previous dv/v studies^{58,59}. The use of

these methods for DAS data under various signal-to-noise levels remains further investigation to reduce the uncertainty.

- The increase of dv/v with frequency is also a characteristic of the exponentially decaying sensitivity of the waves with depth. You may refer to Obermann et al. (2013); 2016 for the body waves and coda examples, and in Yuan et al. (2021), there is a similar case for the surface waves (early coda or late ballistic). Mordret et al. (2020) is an interesting approach for arrays like DAS.

Reference:

Yuan C, Bryan J, Denolle M. Numerical comparison of time-, frequency- and wavelet-domain methods for coda wave interferometry. *Geophysical Journal International*. 2021 Aug;226(2):828-46.

Obermann A, Planès T, Larose E, Campillo M. Imaging preeruptive and coeruptive structural and mechanical changes of a volcano with ambient seismic noise. *Journal of Geophysical Research: Solid Earth*. 2013 Dec;118(12):6285-94.

Obermann A, Planès T, Hadziioannou C, Campillo M. Lapse-time-dependent coda-wave depth sensitivity to local velocity perturbations in 3-D heterogeneous elastic media. *Geophysical Journal International*. 2016 Oct 1;207(1):59-66

Mordret A, Courbis R, Brenguier F, Chmiel M, Garambois S, Mao S, Boué P, Campman X, Lecocq T, Van der Veen W, Hollis D. Noise-based ballistic wave passive seismic monitoring—Part 2: surface waves. *Geophysical Journal International*. 2020 Apr;221(1):692-705.

Reply: Thanks for the suggestion. We added an explanatory sentence in Methods Section 1.2 and cited them properly.

L285-288

The observed dv/v increases with frequency, reflecting an exponentially decaying sensitivity of surface waves with depth^{55,56,58,59} and suggesting that the major contribution comes from the top 20 meters (Supplementary Fig. 1).

- It seems a bit of a missed opportunity not to do the depth inversion, given that there is a very nice velocity model. That said, the study is still excellent!
- While I am not a hydrologist, I find the sophistication of the model for soil moisture excellent and a leap forward in the field of environmental seismology.

Reply: Thank you for the encouraging comments. Regarding the time-lapse imaging, we refer to our reply to your second main comment for more details.

- Inversion of dv/v . Could you summarize in the method, section 3, what is inverted. I understand how to get from soil type to bulk and shear moduli, then to shear wave speed. Soil moisture is also predicted from precipitation, temp, and SMAP. What is the relation that predicts water loss from dv/v ?

Reply: Sorry about the confusion. In Section 3, we use the Solazzi model to obtain dv/v predictions given a daily vadose zone soil moisture change (Δs_0), and to grid search the best fitting Δs_0 that minimizes the misfit between observed and predicted dv/v over seven different frequency bands. Section 4 describes how the soil moisture change can be hydrologically modeled using precipitation, temperature and SMAP data. We added a flowchart in Supp. Fig. 4, as also suggested by Reviewer#1, to show the major steps and revised the text accordingly. We refer to our reply to your first main comment for more details.

Data and code availability:

The team mentioned a GPU-enabled cross correlation workflow, they could/should share their codes. I also recommend writing the output file as a CSV instead of MAT since the data output is a table (time, space, dv/v) to be more standard with community use.

Reply: Thank you for the suggestion. We made the GPU-based cross-correlation code publicly available at GitHub and updated the output data to CSV format to meet the standard for community use in the shared data link. We revised the Data and Code availability section accordingly.

L436-446

The cross-correlation product and multi-frequency dv/v map produced by this study is available at https://data.caltech.edu/records/g4hbc-0mt45?token=eyJhbGciOiJIUzUxMiJ9.eyJpZCI6Ijg1MGFiNmI0LWNjNmEtNDgzMCIhM2Q2LTgwMTUzZWl5NDZhNyIsImRhdGEiOiJ9LCJyYW5kb20iOiJzNTRkZGllODU5MjBhOTkxZjE3MDY2MWE2YWQ5MjFlNiJ9.dsJropZBGFAPLi8VqhXLZfttg_x6BqTixHtV9UJ-QJIsQ-AWOUmxHqPYVH9RffJsPuz4xjsTKNMrwIH5RUf2Q (restricted access for review process, but

will be publicly available at the time of publication).

https://data.caltech.edu/records/g4hbc-0mt45?token=eyJhbGciOiJIUzUxMiJ9.eyJpZCI6Ijg1MGFiNmI0LWNjNmEtNDgzMCIhM2Q2LTgwMTUzZWl5NDZhNyIsImRhdGEiOiJ9LCJyYW5kb20iOiJzNTRkZGllODU5MjBhOTkxZjE3MDY2MWE2YWQ5MjFlNiJ9.dsJropZBGFAPLi8VqhXLZfttg_x6BqTixHtV9UJ-QJIsQ-AWOUmxHqPYVH9RffJsPuz4xjsTKNMrwIH5RUf2Q

https://data.caltech.edu/records/g4hbc-0mt45?token=eyJhbGciOiJIUzUxMiJ9.eyJpZCI6Ijg1MGFiNmI0LWNjNmEtNDgzMCIhM2Q2LTgwMTUzZWl5NDZhNyIsImRhdGEiOiJ9LCJyYW5kb20iOiJzNTRkZGllODU5MjBhOTkxZjE3MDY2MWE2YWQ5MjFlNiJ9.dsJropZBGFAPLi8VqhXLZfttg_x6BqTixHtV9UJ-QJIsQ-AWOUmxHqPYVH9RffJsPuz4xjsTKNMrwIH5RUf2Q

https://data.caltech.edu/records/g4hbc-0mt45?token=eyJhbGciOiJIUzUxMiJ9.eyJpZCI6Ijg1MGFiNmI0LWNjNmEtNDgzMCIhM2Q2LTgwMTUzZWl5NDZhNyIsImRhdGEiOiJ9LCJyYW5kb20iOiJzNTRkZGllODU5MjBhOTkxZjE3MDY2MWE2YWQ5MjFlNiJ9.dsJropZBGFAPLi8VqhXLZfttg_x6BqTixHtV9UJ-QJIsQ-AWOUmxHqPYVH9RffJsPuz4xjsTKNMrwIH5RUf2Q

https://data.caltech.edu/records/g4hbc-0mt45?token=eyJhbGciOiJIUzUxMiJ9.eyJpZCI6Ijg1MGFiNmI0LWNjNmEtNDgzMCIhM2Q2LTgwMTUzZWl5NDZhNyIsImRhdGEiOiJ9LCJyYW5kb20iOiJzNTRkZGllODU5MjBhOTkxZjE3MDY2MWE2YWQ5MjFlNiJ9.dsJropZBGFAPLi8VqhXLZfttg_x6BqTixHtV9UJ-QJIsQ-AWOUmxHqPYVH9RffJsPuz4xjsTKNMrwIH5RUf2Q

L453-454

The GPU-based ambient noise cross-correlation code can be downloaded from https://github.com/zhichaoshen40/DAS_CC_GPU.git.

REVIEWERS' COMMENTS

Reviewer #3 (Remarks to the Author):

The authors did a great job at addressing the reviewers comments